# The Fungal Metabolite (+)-Terrein Abrogates Inflammatory Bone Resorption via the Suppression of TNF-α Production in a Ligature-Induced Periodontitis Mouse Model

**DOI:** 10.3390/jof9030314

**Published:** 2023-03-03

**Authors:** Hidefumi Sako, Kazuhiro Omori, Masaaki Nakayama, Hiroki Mandai, Hidetaka Ideguchi, Saki Yoshimura-Nakagawa, Kyosuke Sakaida, Chiaki Nagata-Kamei, Hiroya Kobayashi, Satoki Ishii, Mitsuaki Ono, Soichiro Ibaragi, Tadashi Yamamoto, Seiji Suga, Shogo Takashiba

**Affiliations:** 1Department of Periodontics and Endodontics, Division of Dentistry, Okayama University Hospital, Okayama 700-8558, Japan; 2Department of Pathophysiology-Periodontal Science, Graduate School of Medicine, Dentistry and Pharmaceutical Sciences, Okayama University, Okayama 700-8525, Japan; 3Department of Oral Microbiology, Graduate School of Medicine, Dentistry and Pharmaceutical Sciences, Okayama University, Okayama 700-8525, Japan; 4Department of Pharmacy, Faculty of Pharmacy, Gifu University of Medical Science, Gifu 509-0261, Japan; 5Division of Applied Chemistry, Graduate School of Natural Sciences and Technology, Okayama University, Okayama 700-8530, Japan; 6Department of Molecular Biology and Biochemistry, Graduate School of Medicine, Dentistry and Pharmaceutical Sciences, Okayama University, Okayama 700-8558, Japan; 7Department of Oral and Maxillofacial Surgery and Biopathology, Graduate School of Medicine, Dentistry and Pharmaceutical Sciences, Okayama University, Okayama 700-8525, Japan

**Keywords:** synthetic (+)-terrein, periodontitis, TNF-α

## Abstract

Current periodontal treatment focuses on the mechanical removal of the source of infection, such as bacteria and their products, and there is no approach to control the host inflammatory response that leads to tissue destruction. In order to control periodontal inflammation, we have previously reported the optimization of (+)-terrein synthesis methods and the inhibitory effect of (+)-terrein on osteoclast differentiation in vitro. However, the pharmacological effect of (+)-terrein in vivo in the periodontitis model is still unknown. In this study, we investigated the effect of synthetic (+)-terrein on inflammatory bone resorption using a ligature-induced periodontitis mouse model. Synthetic (+)-terrein (30 mg/kg) was administered intraperitoneally twice a week to the mouse periodontitis model. The control group was treated with phosphate buffer. One to two weeks after the induction of periodontitis, the periodontal tissues were harvested for radiological evaluation (micro-CT), histological evaluation (HE staining and TRAP staining), and the evaluation of inflammatory cytokine production in the periodontal tissues and serum (quantitative reverse-transcription PCR, ELISA). The synthetic (+)-terrein-treated group suppressed alveolar bone resorption and the number of osteoclasts in the periodontal tissues compared to the control group (*p* < 0.05). In addition, synthetic (+)-terrein significantly suppressed both mRNA expression of TNF-α in the periodontal tissues and the serum concentration of TNF-α (both *p* < 0.05). In conclusion, we have demonstrated that synthetic (+)-terrein abrogates alveolar bone resorption via the suppression of TNF-α production and osteoclast differentiation in vivo. Therefore, we could expect potential clinical effects when using (+)-terrein on inflammatory bone resorption, including periodontitis.

## 1. Introduction

Periodontal disease is a chronic inflammatory disease caused by bacterial infection and biofilm dysbiosis in the oral cavity [1,2]. Tissue destruction, such as alveolar bone resorption, occurs when the host immune response to periodontopathogenic bacterial infections, such as *Porphyromonas gingivalis*, is exaggerated in the oral cavity [3]. In addition, bacteria and their products induce the release of host proteolytic enzymes and cytokines that induce the destruction of periodontal structures [4]. Therefore, current periodontal treatment focuses on the mechanical removal of the bacterial biofilm, mainly by scaling and root planing against periodontopathogenic bacterial infection. The inflammatory response to bacterial biofilm is host-dependent. The outcome of the mechanical removal of bacterial biofilm on tissue healing may also be host-dependent. Therefore, a new therapeutic approach targeting the inflammatory response causing tissue destruction is needed [5].

As the bacterial infection progresses, neutrophils migrate to the gingival sulcus [6], and then immune cells, such as macrophages, dendritic cells, and T cells, infiltrate the connective tissue [7]. These immune cells produce a variety of inflammatory cytokines, including tumor necrosis factor (TNF)-α, interleukin (IL)-1, IL-6, and IL-17 [8]. TNF-α is one of the proinflammatory cytokines produced by a variety of cells, including immune cells, and has been reported to induce osteoclast differentiation [9]. Therefore, it is possible to control the tissue destruction caused by periodontal inflammation by controlling the action of TNF-α. However, there is no approach to these biological inflammatory responses in the current treatment of periodontitis. On the other hand, TNF-α-induced tissue destruction is deeply involved not only in periodontitis but also in the progression of rheumatoid arthritis [10,11]. Molecular target drugs (such as antibody drugs targeting TNF-α; infliximab) have already been marketed and are more effective than the conventional antirheumatic drug methotrexate [12,13]. However, these molecular target drugs block the action of proinflammatory cytokines and strongly suppress the host immune response, so the use of molecular target drugs must pay careful attention to the response of easily infected patients. Therefore, if the effects of proinflammatory cytokines, such as TNF-α, which are involved in the host’s immune response, could be slowly suppressed rather than completely blocked, then this could be applied to the treatment of rheumatoid arthritis and periodontitis, in which inflammatory bone destruction is the main pathology.

(+)-terrein is a low molecular weight (154.16) compound that is isolated as a secondary metabolite from the fungus *Aspergillus terreus* [14]. (+)-terrein, a natural product, has been reported to have a variety of biological effects, including the inhibition of biofilm formation [15], the inhibition of angiogenin secretion in prostate cancer cells [16], and the inhibition of pulp inflammation [17]. Our research group also established an efficient synthesis of (+)-terrein from readily available L-tartrate based on Altenbach’s method [18,19]. Furthermore, the synthesized (+)-terrein was found to have an inhibitory effect on the receptor activator of NF-κB ligand (RANKL) or TNF-α-induced osteoclast differentiation by inhibiting the expression of the nuclear factor of the activated T cell c1 (NFATc1), which is essential for osteoclast differentiation [20]. Based on the above in vitro findings, we hypothesized that (+)-terrein could inhibit inflammatory bone resorption in periodontal tissues if it could effectively suppress TNF-α-induced inflammatory effects and inhibit osteoclast differentiation. However, the anti-inflammatory and antibone resorption effects of (+)-terrein in vivo are still unknown.

In this study, we hypothesized that (+)-terrein can reduce the inflammatory bone resorption in an experimental ligature-induced periodontitis model in mice.

## 2. Materials and Methods

### 2.1. Reagents

(+)-terrein (Figure 1) was synthesized from L-tartaric acid, according to the method described by Mandai et al. [18], and diluted in phosphate-buffered saline (PBS, pH 7.2, Thermo Fisher Scientific, Waltham, MA, USA). All spectra (^1^H and ^13^C nuclear magnetic resonance [NMR] spectra, infrared [IR]) and specific rotation of synthetic (+)-terrein were similar to those of natural (+)-terrein and those previously described [18].

### 2.2. Mouse Ligature-Induced Periodontitis Model

Twelve 10-week-old C57BL/6 wild-type male mice (Clare Corporation, Tokyo, Japan) were used to generate a mouse model of periodontitis using a partial modification of the method [21] (Figure 2). Briefly, mice were subjected to general anesthesia, and then 6–0 silk sutures were ligated to the cervical portion of the maxillary left second molar. The periodontal pathogen, *Porphyromonas gingivalis* W83 strain (1.0 × 10^8^ CFU/mL; 200 μL), was then infiltrated into the silk threads to induce inflammation in the periodontal tissues. In addition, synthetic (+)-terrein (30 mg/kg) was administered intraperitoneally, and PBS was used as a negative control. The time schedule for each procedure (induction of periodontitis, administration of (+)-terrein or PBS, collection of the samples) is shown in Figure 2. Sixteen 10-week-old C57BL/6 wild-type male mice without ligature were used as the control and were treated in the same way as the periodontitis model (administration of (+)-terrein or PBS and collection of the samples).

All animal experiments were conducted under the approval of the Animal Care and Use Committee of Okayama University (OKU-2018748), and the mice were kept under SPF environment. All animal experiments were performed in accordance with the ARRIVE/NC3R guidelines.

### 2.3. Image Analysis Using Microcomputed Tomography (Micro-CT)

To observe alveolar bone resorption, the maxillary bone of each group (N = 3–4) was photographed with a slice width of 6.46 μm using an animal micro-CT imaging device (SkyScan1174v2: Bruker-mCT, Billerica, MA, USA) and reconstructed using the accompanying analysis software (NRecon Bruker-mCT, Bruker). Using Image J software (National Institutes of Health, Bethesda, MD, USA), we quantified the amount of alveolar bone resorption by measuring the distance from the mesial cemento–enamel junction to the alveolar bone apex in 10 sagittal images, including the mesial buccal root of the maxillary left second molar using the method described by Park et al. [22] (Figure 3).

### 2.4. Histological Analysis

Maxillary bone samples from each group (N = 3–4) were fixed in 4% paraformaldehyde solution (pH 7.4: Wako Pure Chemicals, Osaka, Japan) for 1 day. The blocks were immersed in disodium ethylenediaminetetraacetic acid (10% EDTA 2Na solution, pH 7.0, Muto Chemical Co., Ltd., Tokyo, Japan) for 10 days for demineralization and dehydrated in ethanol series. Paraffin-embedded blocks were prepared by making thin sections at 4.5 μm intervals, and paraffin sections were prepared. Subsequently, histopathological analysis was then performed using the following staining methods.

(1)Hematoxylin-Eosin (HE) staining

The prepared paraffin sections were subjected to HE staining according to the normal method and then embedded with coverslip and Mount-Quick (Daido Sangyo Co., Ltd., Tokyo, Japan). Histological images were then observed under a light microscope (BX-50: Olympus, Osaka, Japan).

(2)Tartrate-resistant acid phosphatase (TRAP) staining

The prepared paraffin sections were stained with TRAP staining solution (Cosmo Bio Co., Ltd., Tokyo, Japan) according to the conventional method and then coverslipped with Mount-Quick Aqueous (Daido Sangyo Co., Ltd., Tokyo, Japan). The histological images were then examined by a light microscope. The number of TRAP-positive cells that were in contact with the surface layer of the alveolar bone between the buccal distal root of the left first molar and the buccal mesial root of the left second molar in the maxilla and that had at least three nuclei were counted as osteoclasts. Two sections from each mouse were randomly selected, and osteoclast counts were measured (magnification: ×200) on a total of six sections from three mice in each group.

(3)Immunohistochemistry (IHC) staining

Prepared paraffin sections were deparaffinized and rehydrated, and IHC stained using the avidin-biotin-peroxidase system with the VECTASTAIN Elite ABC Rat Kit (Vector Laboratories, Burlingame, CA, USA). Tissue sections were immersed in 0.3% hydrogen peroxide methanol solution for 30 min at room temperature to remove endogenous peroxidase activity and treated with trypsin (Thermo Fisher Scientific) for 15 min for antigen activation. After blocking with normal rabbit serum for 15 min, primary antibodies were added and reacted overnight at 4 °C. For primary antibodies, anti-Ly-6g and 6c antibodies (Rat-poly; BD Biosciences Pharmingen, San Diego, CA, USA), which are specific markers for peripheral neutrophils and monocytes, were used by diluting 100-fold in PBS; they were washed in PBS and rabbit anti-rat biotin-labeled secondary antibodies were added by diluting 200-fold in PBS and reacted for 30 min at 25 °C. The avidin-biotin-labeled enzyme complex was added and reacted for 30 min, followed by the addition of 0.01% 3,3’-diaminobenzidine (DAB; Nakalai Tesque Co., Ltd., Kyoto, Japan) for staining. Finally, the nuclei were counterstained with Mayer’s Hematoxylin (Wako Junyaku), observed after coverslipping with Mount-Quick, and positive cells were measured as neutrophils.

### 2.5. Analysis of TNF-α Gene Related to Inflammatory Bone Resorption in Periodontal Tissue

The palatal gingiva of the maxillary left second molar was removed from each group (N = 5–10), and total RNA was extracted. To stabilize the RNA, it was immediately immersed in an RNAlater (QIAGEN, Hilden, Germany). RNA was then extracted within 1 day using the RNeasy Plus Mini Kit (QIAGEN). The purity and concentration of the extracted RNA were determined using Nano Drop 2000 (Thermo Fisher Scientific), and after confirming that the value of A260/A280 was greater than 1.8, the RNA was diluted to a concentration of 3–5 μg/mL.

(a) Reverse transcription reaction: The extracted RNA was used as a template for the reverse transcription reaction using Super Script IV VILO Master Mix (Invitrogen, Carlsbad, CA, USA), a cDNA synthesis reaction reagent. Oligo dT primers were annealed by mixing 16 μL of RNA concentration-adjusted solution with 4 μL of Super Script IV VILO Master Mix and heat treated at 25 °C for 10 min. The cDNA was then synthesized by a reverse transcription reaction at 50 °C for 10 min. The reverse transcriptase was then inactivated by heating at 85 °C for 5 min.

(b) Real-time reverse transcription-polymerase chain reaction (RT-PCR): Gene expression of TNF-α was analyzed in periodontal tissues. Primer sequences were designed for the amplification of TNF-α [23] and glyceraldehyde-3-phosphate dehydrogenase (GAPDH) [23] genes, respectively (*Gapdh*: Fw:5′-CACCATGGAGAAGGCCGGGG-3′, Rev:5′-GACGGACACATTGGGGGTAG-3′, bp: 418; *Tnf-α:* Fw:5′-CATCTTCTCAAAATTCGAGTGACAA-3′, Rev:5′-TGGGAGTAGACAAGGTACAACCC-3′, bp: 175) and synthesized according to previous reports. Two microliters of the synthesized cDNA solution were mixed with 0.5 μL each of PCR primers prepared at 1 μM and 10 μL of POWER SYBR Green Master Mix (Thermo Fisher Scientific), and 7 μL of RNase-free water. Fifty cycles of thermal denaturation at 95°C for 15 s and annealing and elongation reactions at 60 °C for 1 min were performed using the 7300 Real-Time PCR System (Thermo Fisher Scientific). Fluorescence of the PCR products generated by this reaction was measured by SDS vLX using RQ Software (Thermo Fisher Scientific). RNA expression levels were quantified by the comparative Ct method [24] using GAPDH as an endogenous control. The relative expression ratios of the ligated side (left side) to the opposite side (right side), where no silk threads were ligated, were calculated, and the gene expression levels were calculated. Gene expression levels less than or equal to 0.5 or greater than 10 on the ligated side were excluded as outliers.

### 2.6. Analysis of TNF-α Level in Serum

Fresh blood was collected from the hearts of each group (N = 3–4), and serum was collected by centrifugation at 875.17× *g* for 15 min at 4 °C. Inflammatory cytokines, TNF-α, which are produced in periodontal inflammatory tissues, were analyzed in serum by enzyme-linked immunosorbent assay (ELISA) using ELISA MAX Deluxe (BioLegend, San Diego, CA, USA) in each well of a microplate. After four washes with PBS-T, the collected serum was diluted 10-fold with PBS and reacted with standards diluted to 4.4, 15.6, 62.5, 250, and 1000 pg/mL for 2 h at room temperature. After four washes with PBS-T, the biotin-conjugated detection antibody was reacted for 1 h at room temperature. For further signal amplification, streptavidin was reacted with PBS-T for 30 min at room temperature after four washes with PBS-T. After five washes with PBS-T, the chromogenic substrate was allowed to react for 15 min at room temperature to develop color. To stop the chromogenic reaction, 100 μL of 2 N sulfuric acid was added, and the absorbance at 450 nm was measured using a microplate reader SH-1000 Lab (Corona Electric Co., Hitachinaka, Japan). All tests were performed in two wells per individual, the serum concentrations of TNF-α were quantified, and the concentrations were set to 0 if the results were below the detection limit.

### 2.7. Biosafety Analysis of Synthetic (+)-Terrein

The biosafety of (+)-terrein was evaluated in mice (mean body weight 26.6 ± 1.5 g, N = 4) treated for 14 days. Weights were measured every 7 days during the treatment period, and mice were euthanized with CO_2_ gas 14 days after treatment. Kidneys and livers, the major metabolizing organs, were removed, and the tissues were fixed in 4% paraformaldehyde solution (pH 7.4) for 1 day. After tissue fixation, the tissues were dehydrated through an ethanol series. Blocks were prepared by paraffin embedding and then thinly sliced at 4.5 μm intervals, and paraffin sections were prepared. Histological changes in the liver and kidney were then observed by HE staining.

### 2.8. Statistical Analysis

We used Q-Q plots to test the normality of data. For statistical analysis, a two-factor analysis of variance (ANOVA) with replication and linear regression was used for multiple comparisons (level of alveolar bone resorption). One-way ANOVA was used for multiple comparisons (number of osteoclasts, gene expression of TNF-α, serum level of TNF-α). The Tukey-Kramer test was used for posthoc analysis. Student’s t-test was used to test the body weight treated with/without (+)-terrein. For statistical analysis, GraphPad Prism8 (GraphPad Software Inc., San Diego, CA, USA) was used. Statistical significance was set at *p* < 0.05.

## 3. Results

### 3.1. Synthetic (+)-Terrein Suppressed Alveolar Bone Resorption in Ligature-Induced Periodontitis Model

There was no statistically significant difference in alveolar bone resorption in the PBS-treated control mice compared to that in the synthetic (+)-terrein-treated control mice, confirming that synthetic (+)-terrein had no adverse effect on healthy alveolar bone (*p* = 0.322; Figure 4a,b). In the periodontitis mice, alveolar bone resorption was significantly increased compared to the control mice, confirming the induction of periodontitis by silk ligation with *Porphyromonas gingivalis* W83 infection (*p* < 0.001, Figure 4a,c). In addition, in the synthetic (+)-terrein-treated periodontitis mice, alveolar bone resorption tended to be suppressed compared to that in the PBS-treated periodontitis mice (*p* = 0.089; Figure 4a,d).

### 3.2. Synthetic (+)-Terrein Suppressed Periodontal Inflammation in Ligature-Induced Periodontitis Model

HE staining revealed no inflammation in the periodontal tissues in both the PBS-treated and synthetic (+)-terrein-treated control mice (Figure 5a), whereas the ligature-induced epithelial depressions, the inflammatory cell infiltration of the subepithelial tissues, and alveolar bone resorption were observed in the PBS-treated periodontitis mice (Figure 5a). IHC staining revealed no neutrophils in both the PBS-treated and synthetic (+)-terrein-treated control mice (Figure 5a), whereas neutrophil infiltration was observed in the PBS-treated periodontitis mice (Figure 5a). The synthetic (+)-terrein-treated periodontitis mice showed reduced neutrophil infiltration compared to that in the PBS-treated periodontitis mice (Figure 5a); TRAP staining showed no osteoclasts in both the PBS-treated and synthetic (+)-terrein-treated control mice (Figure 5a,b), and the PBS-treated periodontitis mice showed a significant increase in the number of osteoclasts compared to that in the PBS-treated and synthetic (+)-terrein-treated control mice (both *p* < 0.001, Figure 5a,b). The synthetic (+)-terrein-treated periodontitis mice showed a significant reduction in the number of osteoclasts compared to that in the PBS-treated periodontitis mice (*p* = 0.04; Figure 5a,b).

### 3.3. Synthetic (+)-Terrein Suppressed TNF-α Gene Expression of Periodontium and Serum TNF-α Levels in a Ligature-Induced Periodontitis Model

In the PBS-treated mice, TNF-α mRNA expression was significantly upregulated in the periodontitis-induced (ligated) area compared with that in the nonligated area (*p* = 0.01; Figure 6a). In the synthetic (+)-terrein-treated mice, the expression of TNF-α mRNA in the periodontitis-induced area was significantly decreased in the PBS-treated mice (*p* = 0.005; Figure 6a).

On the other hand, serum TNF-α was not found in either the PBS-treated or the synthetic (+)-terrein-treated control mice (Figure 6b). In the PBS-treated periodontitis mice, the serum TNF-α levels were significantly increased (*p* = 0.02; Figure 6b). In the periodontitis mice treated with synthetic (+)-terrein, the serum TNF-α levels were significantly reduced down to the uninduced periodontitis level (*p* = 0.02; Figure 6b).

### 3.4. Synthetic (+)-Terrein Had No Toxicity in Mice

The synthetic (+)-terrein-treated control mice did not show any apparent changes in body weight after 7 or 14 days of the experiment compared to the PBS-treated control mice (Figure 7a). In the HE stains of the liver and kidney, the synthetic (+)-terrein-treated control mice did not show any abnormalities in the liver or kidneys, such as congestion and necrosis of the hepatocytes and the inflammation of glomeruli and necrosis of the tubules in the kidneys when compared with the PBS-treated control mice. (Figure 7b).

## 4. Discussion

In this study, we demonstrated that synthetic (+)-terrein suppressed (1) alveolar bone resorption, (2) osteoclastogenesis, and (3) TNF-α expression in the periodontal tissues and serum in a mouse with periodontitis.

A previous study (in vitro) reported that synthetic (+)-terrein inhibited RANKL or TNF-α-induced osteoclast differentiation [20]. The present in vivo study suggests that one of the mechanisms by which synthetic (+)-terrein suppressed alveolar bone resorption in periodontitis mice is the synthetic (+)-terrein-mediated suppression of osteoclast differentiation in the periodontitis tissues, resulting in the inhibition of alveolar bone resorption (Figure 4 and Figure 5b). In addition, synthetic (+)-terrein inhibited neutrophil infiltration into the periodontal tissues of periodontitis mice (Figure 5a). Subsequently, synthetic (+)-terrein reduced the serum TNF-α levels and the expression of TNF-α mRNA in the periodontal tissues (Figure 6). Previous studies have reported that TNF-α activated neutrophils [25] and promoted osteoclast differentiation [26]. One of the effects of synthetic (+)-terrein on the immune response is suggested to be that it inhibits the production of TNF-α in periodontal tissues and induces a series of anti-inflammatory effects, such as neutrophil infiltration and the inhibition of osteoclast formation. The limitation of this study is that we found no statistically significant difference in the amount of alveolar bone resorption between the PBS-treated periodontitis mice and the (+)-terrein-treated periodontitis mice (*p* = 0.089; Figure 4d). However, the *p*-value was close to the 0.05 threshold, and alveolar bone resorption in the (+)-terrein-treated periodontitis group was, on average, 68 μm lower than that in the PBS-treated periodontitis group. This can be interpreted to mean that (+)-terrein may still reduce the rate of alveolar bone resorption, but the results are still inconclusive. This is most likely due to the small sample size and the low number of alveolar bone resorption measurements per animal. In addition, we used *Porphyromonas gingivalis*-infected ligature-induced C57BL/6 mice as a periodontitis model. C57BL/6 mice have been reported to be more resistant to *Porphyromonas gingivalis* infection than BALB/c mice [27,28]. We plan to investigate the anti-inflammatory effect of (+)-terrein in a transgenic mouse model of another inflammatory disease as the same strain, and we believe that it is necessary to use C57BL/6 mice to investigate the anti-inflammatory effect of (+)-terrein.

In this study, there was no difference in body weight change between the synthetic (+)-terrein and PBS groups, and no apparent abnormalities in the kidney or liver histology were observed, suggesting that synthetic (+)-terrein may have few side effects. Furthermore, our results suggest that synthetic (+)-terrein may inhibit inflammatory bone resorption by a mechanism that is different from molecular target drugs, such as antibody drugs. In addition, (+)-terrein can be easily synthesized by organic synthesis, and it will allow us to access a number of its derivatives that show more effective biological activity than original (+)-terrein. Therefore, it would be of great social significance if we could develop therapeutic agents based on synthetic (+)-terrein that can be synthesized by organic synthesis.

In order to investigate the clinical application of synthetic (+)-terrein, it is necessary to elucidate the direct targets of (+)-terrein. We have not been able to identify which part of the TNF-α signaling pathway (+)-terrein acts on when compared to other TNF-α-targeting drugs, such as neutralizing antibodies to TNF-α. In the future, we would like to investigate the molecular targets on which (+)-terrein acts directly. In addition, the duration of the effect after synthetic (+)-terrein administration is unknown, and the effect of synthetic (+)-terrein on the immune response over time and the duration of synthetic (+)-terrein administration needs to be studied in more detail. If the novel mechanism of synthetic (+)-terrein can be elucidated, the potential of synthetic (+)-terrein as a therapeutic agent for inflammatory bone resorption will be further enhanced.

In conclusion, this study demonstrated that synthetic (+)-terrein reduced osteoclast formation and alveolar bone resorption by interfering with TNF-α production in a mouse ligature-induced periodontitis model. The results of this study have insights into the potential clinical use of synthetic (+)-terrein as an antiresorptive agent for the treatment of osteolytic bone diseases, including periodontitis.

## Figures and Tables

**Figure 1 jof-09-00314-f001:**
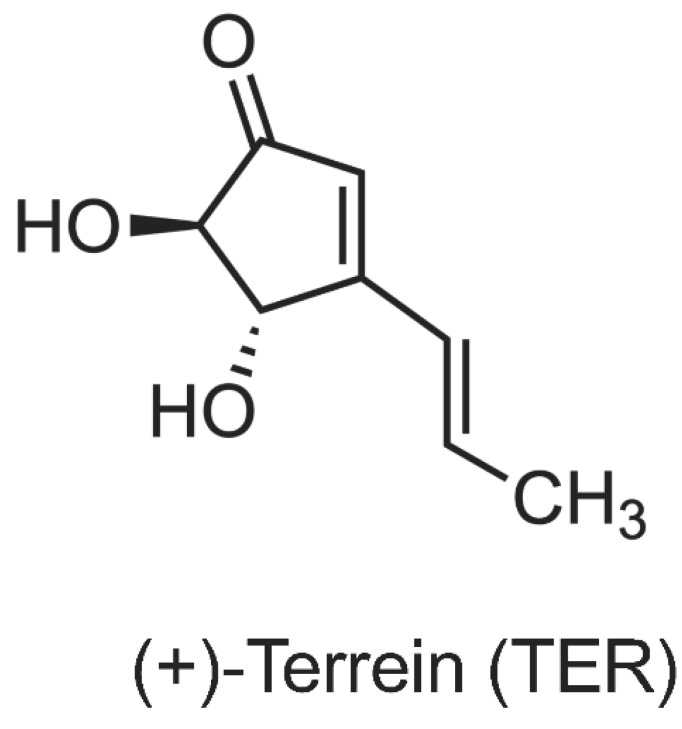
Chemical structure of (+)-terrein.

**Figure 2 jof-09-00314-f002:**
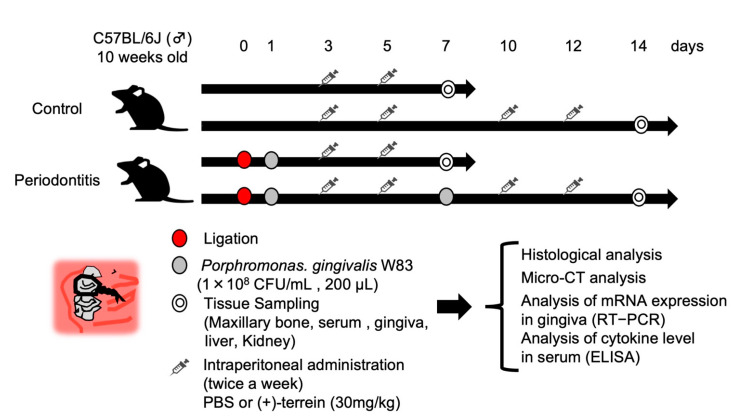
Schedule for animal experiments. 6–0 silk threads were ligated into the maxillary left second molar cervix of 10-week-old C57BL/6 wild-type male mice, and the silk threads were further infiltrated with the bacterial solution of *Porphyromonas gingivalis* W83 strain to induce inflammation in the periodontal tissues. After 7 days of periodontitis induction, the anti-inflammatory effect of synthetic (+)-terrein was observed, and after 14 days, the inhibitory effect of synthetic (+)-terrein on alveolar bone resorption and biosafety was observed. Nonligature mice were used as the control.

**Figure 3 jof-09-00314-f003:**
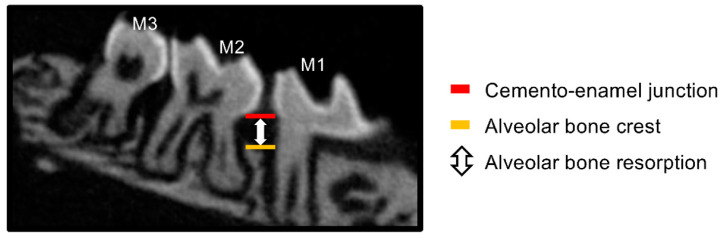
Quantitative method of alveolar bone resorption on micro-CT. Alveolar bone resorption was quantified by measuring the distance from the mesial cemento–enamel junction to the alveolar crest in the sagittal section of the maxillary left second molar on the micro-CT image. M1: maxillary left first molar, M2: maxillary left second molar, M3: maxillary left third molar.

**Figure 4 jof-09-00314-f004:**
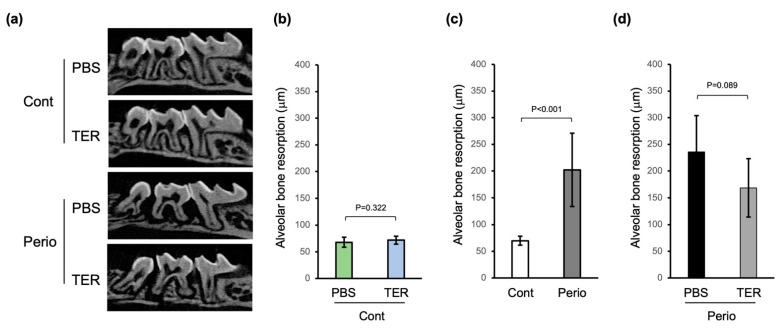
Synthetic (+)-terrein suppressed alveolar bone resorption in a mouse periodontitis model. (**a**) Bone resorption images 14 days after periodontitis induction (sagittal section). Representative micro-CT images for each group are shown. (**b**) The amount of bone resorption (mean ± standard deviation: SD) in PBS-treated or (+)-terrein-treated control mice (N = 8 each) is shown as the length between the cemento–enamel junction (CEJ) and the alveolar bone crest (ABC). (**c**) The amount of bone resorption (mean ± SD) in control mice (N = 16) and ligature-induced periodontitis mice (N = 12) is shown as the length between CEJ and ABC. (**d**) The amount of bone resorption (mean ± SD) in PBS-treated or (+)-terrein-treated ligature-induced periodontitis mice (N = 6 each) are shown as the length between CEJ and ABC. Error bars: SD. PBS: treated with phosphate buffered saline (PBS); TER: treated with (+)-terrein; Cont: nonligature mice; Perio: ligature-induced periodontitis mice. Two-factor ANOVA with replication and linear regression was used for statistical analysis.

**Figure 5 jof-09-00314-f005:**
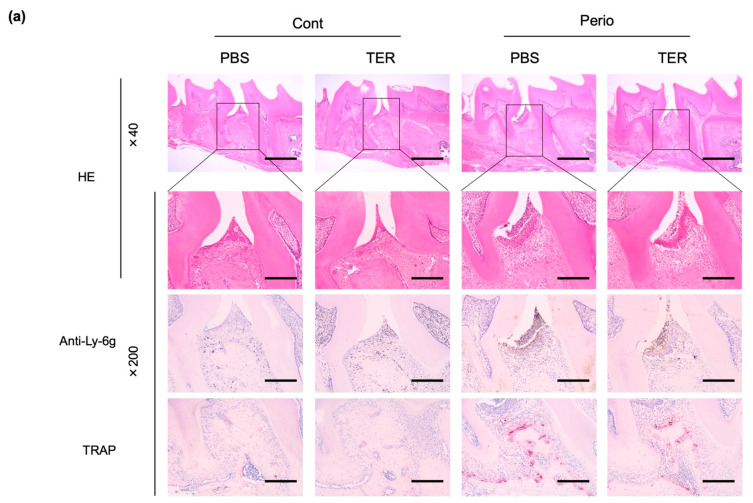
Synthetic (+)-terrein suppressed periodontal inflammation in a mouse periodontitis model. (**a**) Representative examples of HE-stained, IHC (anti-Ly-6g antibody: neutrophils), and TRAP-stained (osteoclasts) images in each group are shown (N = 3–4). The magnified image of the area enclosed by the black line frame of the HE-stained image is shown in the bottom row. Numbers next to the images indicate the magnification; scale bar: 500 µm (at 40× magnification) and 200 µm (at 200× magnification); (**b**) Comparison of the number of osteoclasts. The number of osteoclasts on the proximal alveolar bone surface of left maxillary second molars was measured (N = 3–4, mean ± standard deviation: SD) by defining osteoclasts as cells with positive TRAP staining and more than three nuclei. Error bars: SD, dots: measured values; PBS: treated with phosphate-buffered saline (PBS); TER: treated with (+)-terrein; Cont: nonligature mice; Perio: ligature-induced periodontitis mice. ANOVA/Tukey-Kramer test was used for statistical analysis.

**Figure 6 jof-09-00314-f006:**
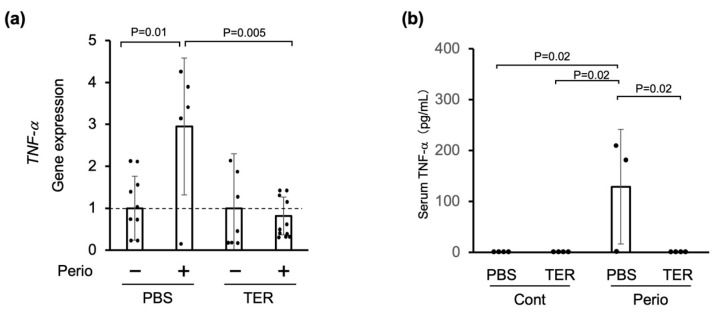
Synthetic (+)-terrein suppressed periodontal TNF-α gene expression and serum TNF-α levels in a ligature-induced periodontitis model. (**a**) Gene expression was analyzed using total RNA extracted from the gingiva (RT-PCR). The level of gene expression on the nonligated side was used as a negative control, and that on the silk-ligated side in the same individual was shown as a relative value (N = 5–10, mean ± standard deviation: SD). Error bars: SD; dots: measured values; dotted lines: control for gene expression levels. PBS: Phosphate-buffered saline (PBS)-treated mice, TER: (+)-terrein-treated mice. (**b**) Serum TNF-α levels were quantified by ELISA (N = 3–4, mean ± SD). Error bars: SD; dots: measured values; PBS: treated with PBS; TER: treated with (+)-terrein; Cont: nonligature mice; Perio: ligature-induced periodontitis mice. ANOVA/Tukey-Kramer test was used for statistical analysis.

**Figure 7 jof-09-00314-f007:**
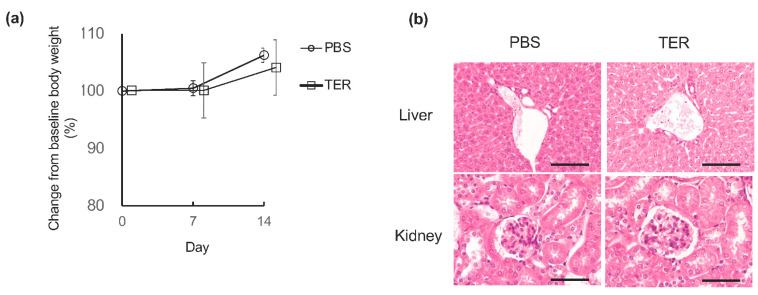
Synthetic (+)-terrein showed no toxicity. (**a**) Effect on body weight. Changes in the body weight of control mice from baseline to 7 and 14 days after the start of the experiment were observed as relative values (mean ± standard deviation: SD) of the baseline body weight (100%) at the start of the experiment (N = 4). Error bars: SD. (**b**) Effects on the liver and kidney. Histological and morphological changes were observed by hematoxylin and eosin (HE) staining of livers and kidneys harvested 14 days after the start of the experiment in control mice. A representative image (N = 4) is shown. Scale bar: 200 µm (at 200× magnification) PBS: control mice treated with phosphate-buffered saline (PBS); TER: control mice treated with synthetic (+)-terrein. Student’s *t*-test was used for statistical analysis.

## Data Availability

The data that support this study are available from the corresponding author upon reasonable request.

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
