# Peer review of "The Fungal Metabolite (+)-Terrein Abrogates Inflammatory Bone Resorption via the Suppression of TNF-α Production in a Ligature-Induced Periodontitis Mouse Model"

_jof, 2023, doi:10.3390/jof9030314_

Round 1

Reviewer 1 Report

Dear authors,

I have read your manuscript titled: The fungal metabolite (+)-terrein abbrogates inflammatory bone resorption via suppression of TNF-a production in mouse ligature-induced periodontitis model. This study is extention of your previously published work on (+)-terrein (doi: 10.1016/j.bmc.2014.07.047; doi: 10.1016/j.heliyon.2018.e00979; doi: 10.1016/j.intimp.2020.106429). The topic is of relevance for researchers interested in novel approaches to treatment of periodontitis.   

However, there are some issues in the overall presentation of your study that need to be addressed. Below is the list of my comments and recommendations:

TITLE

Please replace „TNF-a“ with „TNF-α“ in the title (Line 3).

INTRODUCTION

1.       Please remove the following sentence (Lines 49-51): „The periodontal pathogenic bacteria include gram-negative anaerobes such as P. gingivalis.“

2.       You state that the mechanical removal of bacterial biofilm is host-dependent with respect to the inflammatory response“ (Lines 51-53). What exactly do you mean by that? The sentence is unlcear and it should be rephrased. The inflammatory response to bacterial biofilm is host-dependent. The outcome of the mechanical removal of bacterial biofilm on tissue on tissue healing can also be host-dependent.

3.       Line 59 – instead of „TNF-a“ use „TNF-α“ – make these corrections in the entire manuscript text

4.       You stated that TNF-α is indispensable for the differentiation and activation of RANKL-induced osteoclastogenesis (Lines 59-62). Differentiation and activation should refer to a cell type (osteoclasts) which in turn promote osteoclastogenesis. Please rephrase the statement to be more precise.

5.       You state that the modulation of host inflammatory response is currently not addressed in standard approach to treatment of periodontitis (Lines 64-65; similar statement in Abstract – Lines 25-26). In principle, this is not correct because mechanical removal of bacterial biofilm aims to modulate host inflammatory response. However, there are studies on the effect of different classes of molecules which have multiple physiological roles, including those related to modulation of inflammatory response. Take for example different classes of extracellar matrix components, which in general can promote/inhibit inflammatory response and regeneration of tissue. I would recommend to rephrase the statement (both in Introduction and Abstract) and refer to several interesting papers on the topic (doi: 10.3389/fimmu.2017.01395; doi: 10.1016/j.heliyon.2018.e00719; doi: 10.3390/bioengineering9100566).

6.       It might be better to formulate the closing statement of Introduction section (Lines 90-92) as hypothesis rather than definitive conclusion of the study: “We hypothesized that (+)-terrein can reduce the inflammatory bone resorption in experimental ligature-induced model of periodontitis in mice.”          

MATERIALS AND METHODS

1.       Although you have stated the sample size (lines 126-130), some clarification with regard to that is still needed. In subsection 2.2. Mouse ligature-induced periodontitis model, please specify the total number of mice used for the purpose of study. You also need to clearly specify the number of mice in control group and ligature-induced periodontitis group, followed by the exact number of periodontitis mice used to investigate the anti-inflamatory effect of (+)-terrein, the inhibitory effect of (+)-terrein on alveolar bone resorption, and biosafety of (+)-terrein. State the numbers in this form „(n = ...)“ and insert them in text at appropriate places. It should also be clarified in which cases you used split-mouth design for comparison of relevant parameters – was it just for comparison of gene expression levels and serum concentration of TNF-α? Also, did you analyze only serum concentration of TNF-α or TNF-α and other cytokines by ELISA? (line  

2.       In subsection 2.4. Histological analysis (lines 178-179) you stated that „the number of osteoclasts was measured from six sections from three mice in each group, and two sections  from each mouse were selected at random“. Nobody can verify whether you chose sections at random, and even if you did, why choosing only two sections at random and not all six? You should also state under under what magnification you counted osteoclasts.

3.       In subsection 2.8. Statistical analysis (line 257) you stated that „The data obtained using Shapiro-Wilk test were checked for normality“. The statement should be rephrased because Shapiro-Wilk test is not used for obtaining the data, but to chek the if the distribution of obtained data is normal. If that is the case, you then proceed with parametric tests for hypothesis testing (t-test, ANOVA), and if not, then you use non-parametric tests.

4.       In subsection 2.8. Statistical analysis (lines 257-259) you stated that „multiple comparisons test were performed using one-way analysis of variance (one-way ANOVA). In addition, the Tukey-Kramer test was used for intergroup comparisons“. You must be precise – ANOVA is generally used for simultaneous comparison of multiple groups (3 and more) even though it can also be used for comparison of two groups. In ANOVA, both inter-group (between) and intra-group (within) comparisons are made. Tukey-Kramer is post-hoc test, and we apply Tukey-Kramer (or any other post-hoc test) to define the exact source of statistically significant difference detected by ANOVA. Therefore, to improve the subsection 2.8., you should clearly state what parameters you analyzed with ANOVA and with t-test (level of bone resorption, number of osteoclasts, gene expression of TNF-α , serum concentration of TNF-α etc.). Then simply state that Tukey-Kramer test was used for post-hoc analysis. Since I cannot verify the normality of data distribution, you should add supplementary dataset with descriptive statistics of analyzed parameters (alveolar bone resorption, number of osteoclasts, TNF-α gene expression levels and serum concentrations) and results of Shapiro-Wilk test. For testing of normality of data distribution, I would recommend Q-Q plots instead of Shapiro-Wilk, Kolmogorov-Smirnov and similar tests, since Q-Q plots are graphical presentation of data distribution and are thus more intuitive.

5.       Here is the list of minor issues in Materials and methods:

a)       Line 102 – reference to Fig 2 should be inserted elsewhere in the text, and not in the subsection 2.2. title

b)      Line 142 – add „(micro-CT)“ at the end of the title of subsection 2.3

c)       Line 147 – while it can be used for image processing, ImageJ is basically an image analysis software

d)      Line 150 – please confirm if „buccal root“ is more appropriate term than „cheek root“ and make similar changes in the rest of the manuscript text

e)      Line 154 – replace „proximal cemented enamel border“ with „cemento-enamel border“. It is not clear to me what do you mean by „proximal“? Orientation points on teeth should be „mesial“, „distal“, „coronal“, „apical“, „cervical“...

f)        Line 176 – „lateral root of the cheek of the left first molar“ – please clarify if this refers to „mesio-buccal“, „disto-buccal“ or just „buccal“ root of the upper left first molar

g)       Lines 188-189 – please rephrase the sentence like this: „which are specific markers for peripheral neutrophils and monocytes“

h)      Line 194 – instead of „contrast-stained“ use „counter-stained“

i)        Line 198 – what do you mean by „palatal lateral gingiva“? There is „buccal gingiva“ and „palatal gingiva“

j)        Possibly typo In several places in subsections 2.5. (Line 213), subsection 2.6. (Lines 233, 245) – „TNF- „ instead of „TNF-α“? I've noticed the same typos in Introduction, Results and Discussion sections so please check the entire manuscript and correct them

RESULTS

1.       In reference to point 4 of Materials and methods, I would kindly ask the authors to provide raw data with the output of statistical analysis (ANOVA, t-test) presented in Figures 4, 5 and 6. You may use Excel spreadsheet, and refer to this as Supplementary datasets in figure captions where appropriate. I would generally encourage authors to upload original datasets or at least state the availability of original datasets upon request.

DISCUSSION

1.       In the second paragraph (Lines 376-379) – you state that „synthetic (+)-terrein may inhibit inflammatory bone resorption by a different mechanism from that of drugs targeting specific molecules (e.g., TNF-α).“ What is the importance of that? The ultimate effects of synthetic (+)-terrein and blocking antibody against TNF-α can essentially be the same – inhibition of TNF-α signaling pathway is anti-inflammatory no matter how you achieve it, which in turn may help to reduce inflammatory bone resorption as is the case in periodontal tissue. What you lack here is the comparison with results from studies on the effectiveness of other agents performed on experimental mouse models of periodontitis.

2.       You should comment your results in the context of limitations of ligature-induced periodontitis model compared with mono- or polymicrobial oral lavage periodontitis models. Why did you choose to work on C57BL/6 mice? As I understand, this C57BL/6 strain is resistant to periodontitis even under high exposure to P. gingivalis compared to BALB/c mice (please check doi: 10.1128/IAI.68.6.3103-3107.2000, and doi: 10.1186/1471-2156-14-68).

3.       Please rephrase the statement (Lines 384-385): „To investigate the clinical application of synthetic (+)-terrein, it is necessary to elucidate the direct target molecules and metabolic pathways that remain to be clarified.“ The sentence is repetitive so please remove „that remain to be clarified“ part. What other metabolic pathways do you presume might be affected by synthetic (+)-terrein?

Several „TNF- “ typos (Lines 368, 369, 371, 392).

Author Response

Reviewer #1

I have read your manuscript titled: The fungal metabolite (+)- terrein abbrogates inflammatory bone resorption via suppression of TNF-a production in mouse ligature-induced periodontitis model. This study is extention of your previously published work on (+)-terrein (doi: 10.1016/j.bmc.2014.07.047; doi: 10.1016/j.heliyon.2018.e00979; doi: 10.1016/j.intimp.2020.106429). The topic is of relevance for researchers interested in novel approaches to treatment of periodontitis.

However, there are some issues in the overall presentation of your study that need to be addressed. Below is the list of my comments and recommendations:

Thank you for reviewing the manuscript and providing several useful comments. We have provided point-by-point responses to each comment, and the red text indicates the revised manuscript.

TITLE
Please replace „TNF-a“ with „TNF-α“ in the title (Line 3).

We have corrected the section.

INTRODUCTION

  1. Please remove the following sentence (Lines 49-51): „The periodontal pathogenic bacteria include gram-negative anaerobes such as P. gingivalis.“

The sentence was deleted.

  1. You state that the mechanical removal of bacterial biofilm is host-dependent with respect to the inflammatory response“ (Lines 51-53). What exactly do you mean by that? The sentence is unlcear and it should be rephrased. The inflammatory response to bacterial biofilm is host-dependent. The outcome of the mechanical removal of bacterial biofilm on tissue healing can also be host-dependent.

We have revised the text based on the reviewer’s suggestion, “The inflammatory response to bacterial biofilm is host-dependent. The outcome of the mechanical removal of bacterial biofilm on tissue healing may also be host-dependent”.

  1. Line 59 – instead of „TNF-a“ use „TNF-α“ – make these corrections in the entire manuscript text

We have corrected the section.

  1. You stated that TNF-α is indispensable for the differentiation and activation of RANKL-induced osteoclastogenesis (Lines 59- 62). Differentiation and activation should refer to a cell type (osteoclasts) which in turn promote osteoclastogenesis. Please rephrase the statement to be more precise.

We corrected the relevant text, “TNF-a is one of the pro-inflammatory cytokines produced by a variety of cells, including immune cells, and has been reported to induce osteoclast differentiation”.

  1. You state that the modulation of host inflammatory response is currently not addressed in standard approach to treatment of periodontitis (Lines 64-65; similar statement in Abstract – Lines 25-26). In principle, this is not correct because mechanical removal of bacterial biofilm aims to modulate host inflammatory response. However, there are studies on the effect of different classes of molecules which have multiple physiological roles, including those related to modulation of inflammatory response. Take for example different classes of extracellar matrix components, which in general can promote/inhibit inflammatory response and regeneration of tissue. I would recommend to rephrase the statement (both in Introduction and Abstract) and refer to several interesting papers on the topic (doi: 10.3389/fimmu.2017.01395; doi: 10.1016/j.heliyon.2018.e00719; doi: 10.3390/bioengineering9100566).

Thank you for your comments. We have revised the abstract and introduction to reflect the reviewer’s comments, citing the references the reviewer advised.

  1. It might be better to formulate the closing statement of Introduction section (Lines 90-92) as hypothesis rather than definitive conclusion of the study: “We hypothesized that (+)- terrein can reduce the inflammatory bone resorption in experimental ligature-induced model of periodontitis in mice.”

Thank you for your advice. We revised the text to reflect the reviewer’s advice.

MATERIALS AND METHODS

  1. Although you have stated the sample size (lines 126-130), some clarification with regard to that is still needed. In subsection 2.2. Mouse ligature-induced periodontitis model, please specify the total number of mice used for the purpose of study. You also need to clearly specify the number of mice in control group and ligature-induced periodontitis group, followed by the exact number of periodontitis mice used to investigate the anti-inflamatory effect of (+)-terrein, the inhibitory effect of (+)- terrein on alveolar bone resorption, and biosafety of (+)-terrein. State the numbers in this form „(n = ...)“ and insert them in text at appropriate places. It should also be clarified in which cases you used split-mouth design for comparison of relevant parameters – was it just for comparison of gene expression levels and serum concentration of TNF-α? Also, did you analyze only serum concentration of TNF-α or TNF-α and other cytokines by ELISA?

As noted by the reviewer, we have included the number of animals used in the text. In addition, silk thread ligation was performed only on the left maxillary second molar in this study. Therefore, the analysis of TNF-a mRNA gene expression in Figure 6(a) uses the unligated right gingiva as a control. There was also an error in the notation of Figure 6(a). The figure legend of 6(a) has also been corrected. In addition, only serum TNF-a was measured in this study because only TNF-a showed a suppressive effect on gene expression with synthetic (+)-terrein.

  1. In subsection 2.4. Histological analysis (lines 178-179) you stated that „the number of osteoclasts was measured from six sections from three mice in each group, and two sections from each mouse were selected at random“. Nobody can verify whether you chose sections at random, and even if you did, why choosing only two sections at random and not all six? You should also state under under what magnification you counted osteoclasts.

We are very sorry. The wording of the sentence was incorrect. We have corrected the sentence as follows and added the observation magnification (x200).

  1. In subsection 2.8. Statistical analysis (line 257) you stated that „The data obtained using Shapiro-Wilk test were checked for normality“. The statement should be rephrased because Shapiro-Wilk test is not used for obtaining the data, but to check the if the distribution of obtained data is normal. If that is the case, you then proceed with parametric tests for hypothesis testing (t-test, ANOVA), and if not, then you use non-parametric tests.

The text in 2.8 has been revised in conjunction with the comments in 4.

  1. In subsection 2.8. Statistical analysis (lines 257-259) you stated that „multiple comparisons test were performed using one-way analysis of variance (one-way ANOVA). In addition, the Tukey-Kramer test was used for intergroup comparisons“. You must be precise – ANOVA is generally used for simultaneous comparison of multiple groups (3 and more) even though it can also be used for comparison of two groups. In ANOVA, both inter-group (between) and intra-group (within) comparisons are made. Tukey-Kramer is post-hoc test, and we apply Tukey- Kramer (or any other post-hoc test) to define the exact source of statistically significant difference detected by ANOVA. Therefore, to improve the subsection 2.8., you should clearly state what parameters you analyzed with ANOVA and with t-test (level of bone resorption, number of osteoclasts, gene expression of TNF-α , serum concentration of TNF-α etc.). Then simply state that Tukey-Kramer test was used for post-hoc analysis. Since I cannot verify the normality of data distribution, you should add supplementary dataset with descriptive statistics of analyzed parameters (alveolar bone resorption, number of osteoclasts, TNF-α gene expression levels and serum concentrations) and results of Shapiro-Wilk test. For testing of normality of data distribution, I would recommend Q-Q plots instead of Shapiro- Wilk, Kolmogorov-Smirnov and similar tests, since Q-Q plots are graphical presentation of data distribution and are thus more intuitive.

Following the reviewer’s advice, we have revised the text of 2.8 as follows, “For statistical analysis, one-way analysis of variance (ANOVA) was used for multiple comparisons (level of bone resorption, number of osteoclasts, gene expression of TNF-a, serum level of TNF- a). The Tukey-Kramer test was used for post-hoc analysis. Student's t-test was used to test the body weight treated with/without (+)-terrein. For statistical processing, GraphPad Prism8 (GraphPad Software Inc, San Diego, CA, USA) was used for statistical processing, and P < 0.05 was judged to be significantly different.” We also attached the Excel file including the dataset of Figure 4-7, statistical analysis (ANOVA, Tukey-Kramer test, Student’s t-test, Q-Q plot).

  1. Here is the list of minor issues in Materials and methods:
  2. a) Line 102 – reference to Fig 2 should be inserted elsewhere in the text, and not in the subsection 2.2. title

We have removed Fig. 2 from the title of subsection 2.2.

  1. b) Line 142 – add „(micro-CT)“ at the end of the title of subsection 2.3

We add the text, micro-CT, at the end of the title of subsection 2.3.

  1. c) Line 147 – while it can be used for image processing, ImageJ is basically an image analysis software

We have revised the text of the image analysis software Image J.

  1. d) Line 150 – please confirm if „buccal root“ is more appropriate term than „cheek root“ and make similar changes in the rest of the manuscript text

We have rephrased the “cheek root” to “buccal root” throughout the rest of the manuscript.

  1. e) Line 154 – replace „proximal cemented enamel border“ with „cemento-enamel border“. It is not clear to me what do you mean by „proximal“? Orientation points on teeth should be „mesial“, „distal“, „coronal“, „apical“, „cervical“...

We have rephrased the “cheek root” to “buccal root” throughout the rest of the manuscript.

  1. f) Line 176 – „lateral root of the cheek of the left first molar“ – please clarify if this refers to „mesio- buccal“, „disto-buccal“ or just „buccal“ root of the upper left first molar

We have rephrased the text.

  1. g) Lines 188-189 – please rephrase the sentence like this: „which are specific markers for peripheral neutrophils and monocytes“

We have rephrased the text.

  1. h) Line 194 – instead of „contrast-stained“ use „counter-stained“

We have rephrased the text.

  1. i) Line 198 – what do you mean by „palatal lateral gingiva“? There is „buccal gingiva“ and „palatal gingiva“

We have rephrased the text.

  1. j) Possibly typo In several places in subsections 2.5. (Line 213), subsection 2.6. (Lines 233, 245) – „TNF- „ instead of „TNF-α“? I've noticed the same typos in Introduction, Results and Discussion sections so please check the entire manuscript and correct them

We have rephrased the text throughout the rest of the manuscript.

RESULTS

  1. In reference to point 4 of Materials and methods, I would kindly ask the authors to provide raw data with the output of statistical analysis (ANOVA, t-test) presented in Figures 4, 5 and 6. You may use Excel spreadsheet, and refer to this as Supplementary datasets in figure captions where appropriate. I would generally encourage authors to upload original datasets or at least state the availability of original datasets upon request.

Thank you for your comment, we added the statement of data availability in the manuscript. In addition, we uploaded the Excel file of dataset_Figure 4-7 with the output of statistical analysis.

DISCUSSION

  1. In the second paragraph (Lines 376-379) – you state that „synthetic (+)-terrein may inhibit inflammatory bone resorption by a different mechanism from that of drugs targeting specific molecules (e.g., TNF-α).“ What is the importance of that? The ultimate effects of synthetic (+)-terrein and blocking antibody against TNF-α can essentially be the same – inhibition of TNF-α signaling pathway is anti-inflammatory no matter how you achieve it, which in turn may help to reduce inflammatory bone resorption as is the case in periodontal tissue. What you lack here is the comparison with results from studies on the effectiveness of other agents performed on experimental mouse models of periodontitis.

As the reviewer pointed out, we reported the possibility that synthetic (+)-terrein could inhibit inflammatory alveolar bone destruction by suppressing the TNF-a signaling pathway as well as TNF- a neutralizing antibodies. As the reviewer pointed out, we added as a limitation of this study that a comparison of the effect with other TNF- a therapeutics will be needed in the future.

  1. You should comment your results in the context of limitations of ligature-induced periodontitis model compared with mono- or polymicrobial oral lavage periodontitis models. Why did you choose to work on C57BL/6 mice? As I understand, this C57BL/6 strain is resistant to periodontitis even under high exposure to P. gingivalis compared to BALB/c mice (please check doi: 10.1128/IAI.68.6.3103-3107.2000, and doi: 10.1186/1471-2156-14-68).

As the reviewer mentioned, C57BL/6 mice have been reported to be resistant to P. gingivalis infection. We are planning to investigate the anti-inflammatory effect of (+)-terrein in a transgenic mouse model of another disease, and we believe it is necessary to use mice of the same strain as C57BL/6 mice.

  1. Please rephrase the statement (Lines 384-385): „To investigate the clinical application of synthetic (+)-terrein, it is necessary to elucidate the direct target molecules and metabolic pathways that remain to be clarified.“ The sentence is repetitive so please remove „that remain to be clarified“ part. What other metabolic pathways do you presume might be affected by synthetic (+)-terrein?

This study suggested that synthetic (+)-terrein may inhibit alveolar bone resorption by suppressing TNF-a signaling. However, we were not been able to identify which part of the TNF-a signaling pathway (+)-terrein acts on. In the future, we would like to investigate the molecular targets on which (+)-terrein acts directly.

Several „TNF- “ typos (Lines 368, 369, 371, 392).

We have rephrased the text throughout the rest of the manuscript.

Reviewer 2 Report

In a previous work the authors showed that synthetic (+)-terrein suppressed osteoclast differentiation in vitro, therefore in this study they aimed to show its effect in vivo, using the ligature -induced periodontitis model.

Authors successfully demonstrated that synthetic (+)-terrein prevents alveolar bone resorption, osteoclastogenesis and signs of periodontal inflammation

Specific comments to the authors are detailed below:

Line 47:  update definition of periodontitis, considering that it is a dysbiotic disease

Line 50: replace “gram-negative” by “Gram-negative”

Line 65 and 67: add the missing “a” in TNF-a

Line 102: cite the figure in the main text, not in the subtitle

Line 104: please mention the method in the text in order to make it clearer

Line 107: please indicate the medium used for Porphyromonas gingivalis W83 culture and indicate growing time (exponential phase?) previous to the OD600 adjustment

Line 102-134: there are many experimental groups each one including several time points. The description of each one tends to be a little confusing. In order to make it clearer, I would recommend including all experimental groups in a figure or table (it could be in Figure 2 or a new one).

Line 135-141: please describe in the figure legend all the symbols used in the figure (the syringe is missing).

Author Response

Reviewer 2

In a previous work the authors showed that synthetic (+)-terrein suppressed osteoclast differentiation in vitro, therefore in this study they aimed to show its effect in vivo, using the ligature - induced periodontitis model.

Authors successfully demonstrated that synthetic (+)-terrein prevents alveolar bone resorption, osteoclastogenesis and signs of periodontal inflammation

Thank you for your comments. We have included point-by-point responses to each comment (Q5), and provided the revised manuscript with the revised red text with underline.

Line 47: update definition of periodontitis, considering that it is a dysbiotic disease

We revised introductory text and added references.

Line 50: replace “gram-negative” by “Gram-negative”

At the suggestion of Reviewer#1, this sentence has been removed.

Line 65 and 67: add the missing “a” in TNF-a

We have rephrased the text.

Line 102: cite the figure in the main text, not in the subtitle

We have rephrased the text.

Line 104: please mention the method in the text in order to make it clearer

We have rephrased the text.

Line 107: please indicate the medium used for Porphyromonas gingivalis W83 culture and indicate growing time (exponential phase?) previous to the OD600 adjustment

The text in the relevant section has been corrected.

Line 102-134: there are many experimental groups each one including several time points. The description of each one tends to be a little confusing. In order to make it clearer, I would recommend including all experimental groups in a figure or table (it could be in Figure 2 or a new one).

Thank you for the advice. We changed the description of the conditions for each experiment.

Line 135-141: please describe in the figure legend all the symbols used in the figure (the syringe is missing).

Thank you for your suggestion. We have added a description of the syringe to the image in Figure 2.

Round 2

Reviewer 1 Report

I have read the revised version of the manuscript. The authors have significantly improved the manuscript. It should also be noted that the authors have been transparent when sharing the supplementary data. I am also grateful to the authors for providing me with the detailed account of changes introduced in the manuscript.

However, there are still some important corrections that need to be made before I can recommend the manuscript for publishing.

MAJOR ISSUES

Here, I will focus on analysis of the amount of alveolar bone resorption (ABR) shown on Figure 4. You divided animals in 8 groups and analyzed differences with one-way ANOVA. The problem here is that some groups (negative controls and test group) are extremely small because you additionally divided them by the time-frame (7 days and 14 days, which makes them n= 3 each). The same was done for controls (n = 4 each). Another problem is unequal number of animals in control and test groups (n = 16 vs. n = 12; or n = 4 vs. n = 3). Two-factor ANOVA with replication would be more appropriate for the comparison of ABR between groups, but it cannot be done because of the unequal size of control and test groups.

Anyway, I've re-run the analysis of ABR according to several key points:

1. Does terrein have adverse effects on ABR in non-periodontitis animals? I tested this by two-factor ANOVA. There was no statistically significant difference between CTRL and CTRL+TER groups (Sheet 1), ergo terrein showed no adverse effects. This is in line with your findings.

2. Based on comparison of ABR in controls and test groups, you I have confirmed that you had successfully established periodontitis model. Here, due to unequal size of groups, I used linear regression with dummy variables. I have also aggregated data from control groups (CTRL & CTRL+TER) and test groups (PERIO+PBS & PERIO+TER) without dividing them by time-frame (1 week vs. 2 weeks) (Sheet 2). I have found statistically significant difference ABR between controls and test group.

3. The last point is the effectiveness of terrein on reduction of ABR. I have used linear regression with dummy variable. I have also aggregated all measurements of ABR from negative controls (PERIO+PBS) and test group (PERIO-TER) not dividing them in subgroups based on time-frame (1 week vs. 2 weeks) (Sheet 2). Unlike you, I found no statistically significant difference (P = 0.08) in the amount of ABR between negative controls and PERIO+TER. However, the P-value was close to 0.05 threshold, and ABR in PERIO+TER group was on average 68 μm lower than that in negative controls. This can be interpreted in the way that terrein might still reduce the rate of ABR, but results are still inconclusive. Most likely due to small sample size, and due to small number of measurements of ABR per animal (the latter could also be used to increase the sample size). You should comment on this in the Discussion section under the study limitations.

Based on these points, some changes in presentation of results on Figure 4 need to be introduced – please check the figure I inserted in Sheet 3. On Figure 4, I would keep the panel A (with micro-CT image of ABR), but would change the graphs representing the analysis of ABR in groups.

Additionally, corrections should also be made in the text of Results section (where appropriate) and in sub-section 2.8. of Materials & Methods regarding the selection of statistical tests – apart from one-way ANOVA, now there are two-factor ANOVA with replication and linear regression. You may also feel free to consult your statistics advisor if some of my interventions are unclear.

MINOR ISSUES

1. It is not clear to me – was ligature inserted in all animals? Do you define controls simply as animals with ligature not inflitrated with Porphyromonas gingivalis?

2. Line 127-130 (Materials & Methods; sub-section 2.3) – rephrase the sentence like this: „Using ImageJ sofware (National Institute of Health, Bethesda, MD, USA), we quantified the amount of alveolar bone resorption by measuring the distance from the mesial cemento-enamel junction to the alveolar bone ridge in 10 sagittal images...“

3. Materials & Methods; sub-section 2.8 – state that you used Q-Q plots for testing the normality of data.

4. Line 240-241 (Materials & Methods; sub-section 2.8) – rephrase the sentence like this: „For statistical analysis, GraphPad Prism8 (GraphPad  Software Inc, San Diego, CA, USA) was used. Statistical significance was set at α = 0.05 (P < 0.05)."

5. Be specific about the total number of animals used in the study (is it 28?). In sub-section 2.2 of Materials & Methods (Mouse ligature-induced periodontitis model) rephrase the opening sentence (line 103) stating the total number of animals used: „28 ten-week-old C57BL/6 wild-type male mice...“

6. You still haven't responded regarding my comment about C56BL/6 strain. Why did you choose this strain (which is resistant to periodontitis) for the purpose of your study? Please insert your comment in Discussion section where appropriate.

Author Response

Reviewer #1

I have read the revised version of the manuscript. The authors have significantly improved the manuscript. It should also be noted that the authors have been transparent when sharing the supplementary data. I am also grateful to the authors for providing me with the detailed account of changes introduced in the manuscript.

However, there are still some important corrections that need to be made before I can recommend the manuscript for publishing.

Thank you for reviewing the manuscript and providing statistical analysis support. We have provided point-by-point responses to each comment, and the blue text indicates the revised manuscript.

MAJOR ISSUES

Here, I will focus on analysis of the amount of alveolar bone resorption (ABR) shown on Figure 4. You divided animals in 8 groups and analyzed differences with one-way ANOVA. The problem here is that some groups (negative controls and test group) are extremely small because you additionally divided them by the time-frame (7 days and 14 days, which makes them n= 3 each). The same was done for controls (n = 4 each). Another problem is unequal number of animals in control and test groups (n = 16 vs. n = 12; or n = 4 vs. n = 3). Two-factor ANOVA with replication would be more appropriate for the comparison of ABR between groups, but it cannot be done because of the unequal size of control and test groups.

Anyway, I've re-run the analysis of ABR according to several key points:

  1. Does terrein have adverse effects on ABR in non-periodontitis animals? I tested this by two-factor ANOVA. There was no statistically significant difference between CTRL and CTRL+TER groups (Sheet 1), ergo terrein showed no adverse effects. This is in line with your findings.
  2. Based on comparison of ABR in controls and test groups, you I have confirmed that you had successfully established periodontitis model. Here, due to unequal size of groups, I used linear regression with dummy variables. I have also aggregated data from control groups (CTRL & CTRL+TER) and test groups (PERIO+PBS & PERIO+TER) without dividing them by time- frame (1 week vs. 2 weeks) (Sheet 2). I have found statistically significant difference ABR between controls and test group.
  3. The last point is the effectiveness of terrein on reduction of ABR. I have used linear regression with dummy variable. I have also aggregated all measurements of ABR from negative controls (PERIO+PBS) and test group (PERIO-TER) not dividing them in subgroups based on time-frame (1 week vs. 2 weeks) (Sheet 2). Unlike you, I found no statistically significant difference (P = 0.08) in the amount of ABR between negative controls and PERIO+TER. However, the P-value was close to 0.05 threshold, and ABR in PERIO+TER group was on average 68 μm lower than that in negative controls. This can be interpreted in the way that terrein might still reduce the rate of ABR, but results are still inconclusive. Most likely due to small sample size, and due to small number of measurements of ABR per animal (the latter could also be used to increase the sample size). You should comment on this in the Discussion section under the study limitations.

Based on these points, some changes in presentation of results on Figure 4 need to be introduced – please check the figure I inserted in Sheet 3. On Figure 4, I would keep the panel A (with micro-CT image of ABR), but would change the graphs representing the analysis of ABR in groups.

Additionally, corrections should also be made in the text of Results section (where appropriate) and in sub-section 2.8. of Materials & Methods regarding the selection of statistical tests – apart from one-way ANOVA, now there are two-factor ANOVA with replication and linear regression. You may also feel free to consult your statistics advisor if some of my interventions are unclear.

We appreciate the statistical analysis support provided by reviewer #1. We followed reviewer#1's suggestions and revised the Figure 4 and Figure 4 legends, and the text of the Results and Discussion sections according to the suggestions. We have also revised the description of Figures 5 and 6 following the revision of Figure 4.

MINOR ISSUES

  1. It is not clear to me – was ligature inserted in all animals? Do you define controls simply as animals with ligature not inflitrated with Porphyromonas gingivalis?

We used non-ligated mice as a control in this study. We have added text to the Materials and methods section #2.2 about the control and added the image of control in Figure 2.

  1. Line 127-130 (Materials & Methods; sub-section 2.3) – rephrase the sentence like this: „Using ImageJ sofware (National Institute of Health, Bethesda, MD, USA), we quantified the amount of alveolar bone resorption by measuring the distance from the mesial cemento-enamel junction to the alveolar bone ridge in 10 sagittal images...“

We have rephrased the text.

  1. Materials & Methods; sub-section 2.8 – state that you used Q-Q plots for testing the normality of data.

We have added the text about Q-Q plots in section #2.8.

  1. Line 240-241 (Materials & Methods; sub-section 2.8) – rephrase the sentence like this: „For statistical analysis, GraphPad Prism8 (GraphPad Software Inc, San Diego, CA, USA) was used. Statistical significance was set at α = 0.05 (P < 0.05)."

We have rephrased the text.

  1. Be specific about the total number of animals used in the study (is it 28?). In sub-section 2.2 of Materials & Methods (Mouse ligature-induced periodontitis model) rephrase the opening sentence (line 103) stating the total number of animals used: „28 ten-week-old C57BL/6 wild-type male mice...“

We have rephrased the text on the total number of animals used.

  1. You still haven't responded regarding my comment about C56BL/6 strain. Why did you choose this strain (which is resistant to periodontitis) for the purpose of your study? Please insert your comment in Discussion section where appropriate.

As the reviewer mentioned, C57BL/6 mice have been reported to be resistant to P. gingivalis infection. We are planning to investigate the anti-inflammatory effect of (+)-terrein in a transgenic mouse model of another disease, and we believe it is necessary to use mice of the same strain as C57BL/6 mice. We have added text in the Discussion section about the use of C57BL/6 mice in this study and describe the limitation of this study using C57BL/6 mice.
